# E-Cigarette Exposure Decreases Bone Marrow Hematopoietic Progenitor Cells

**DOI:** 10.3390/cancers12082292

**Published:** 2020-08-14

**Authors:** Gajalakshmi Ramanathan, Brianna Craver-Hoover, Rebecca J. Arechavala, David A. Herman, Jane H. Chen, Hew Yeng Lai, Samantha R. Renusch, Michael T. Kleinman, Angela G. Fleischman

**Affiliations:** 1Division of Hematology/Oncology, Department of Medicine, University of California, Irvine, CA 92697, USA; gramanat@hs.uci.edu; 2Department of Biological Chemistry, University of California, Irvine, CA 92617, USA; bcraver@uci.edu (B.C.-H.); janehc1@uci.edu (J.H.C.); hylai@uci.edu (H.Y.L.); 3Division of Occupational and Environmental Medicine, University of California, Irvine, CA 92617, USA; rebeccj2@uci.edu (R.J.A.); daherman@uci.edu (D.A.H.); srenusch@uci.edu (S.R.R.); mtkleinm@uci.edu (M.T.K.)

**Keywords:** electronic cigarette, hematopoietic stem cell, myeloid progenitors, myeloproliferative neoplasm, lipopolysaccharide

## Abstract

Electronic cigarettes (E-cigs) generate nicotine containing aerosols for inhalation and have emerged as a popular tobacco product among adolescents and young adults, yet little is known about their health effects due to their relatively recent introduction. Few studies have assessed the long-term effects of inhaling E-cigarette smoke or vapor. Here, we show that two months of E-cigarette exposure causes suppression of bone marrow hematopoietic stem and progenitor cells (HSPCs). Specifically, the common myeloid progenitors and granulocyte-macrophage progenitors were decreased in E-cig exposed animals compared to air exposed mice. Competitive reconstitution in bone marrow transplants was not affected by two months of E-cig exposure. When air and E-cig exposed mice were challenged with an inflammatory stimulus using lipopolysaccharide (LPS), competitive fitness between the two groups was not significantly different. However, mice transplanted with bone marrow from E-cigarette plus LPS exposed mice had elevated monocytes in their peripheral blood at five months post-transplant indicating a myeloid bias similar to responses of aged hematopoietic stem cells (HSC) to an acute inflammatory challenge. We also investigated whether E-cigarette exposure enhances the selective advantage of hematopoietic cells with myeloid malignancy associated mutations. E-cigarette exposure for one month slightly increased *JAK2^V617F^* mutant cells in peripheral blood but did not have an impact on *TET2*^−/−^ cells. Altogether, our findings reveal that chronic E-cigarette exposure for two months alters the bone marrow HSPC populations but does not affect HSC reconstitution in primary transplants.

## 1. Introduction

Electronic cigarettes (E-cigarettes or E-cigs) or the electronic nicotine delivery system (ENDS) emerged as a popular alternative to conventional tobacco smoking with currently 10 million adult and over 5 million middle and high school students using E-cigarettes in the US [1,2]. E-cigarettes have evolved rapidly but essentially consist of a battery and heating coil that aerosolizes e-liquid containing the solvents propylene glycol and vegetable glycerin to deliver nicotine for inhalation. The availability of a variety of flavorings in the E-cigarette liquid make it further appealing to youth users. E-cigarette usage increases the risk for nicotine addiction and impacts normal brain development among adolescents and increases the likelihood of smoking tobacco cigarettes, thus adversely reversing progress made in declining youth tobacco use [3,4]. The dramatic increase in E-cigarette popularity among adults and more concerningly in adolescents has led to the urgent need to investigate the health-related effects of E-cigarette smoking or vaping. 

Hematopoietic stem and progenitor cells (HSPCs) play a pivotal role in maintaining the body’s steady-state blood and immune cell populations throughout one’s lifespan and also increasing output in response to hematopoietic stress such as in the presence of an infection [5,6]. Traditional cigarette smoking is known to affect peripheral blood cell counts causing neutrophilia and erythrocytosis [7,8] and has been reported to decrease the number of bone marrow HSPCs [9]. Smoking status has been correlated with the presence of myeloproliferative neoplasms (MPNs) [10,11,12,13], a group of hematologic malignancies arising from clonal outgrowth of a somatically mutated HSC, most commonly in JAK2 (*JAK2^V617F^*) which results in the overproduction of mature cells of the myeloid lineage [14,15,16,17]. Smoking status is also correlated with the risk of clonal hematopoiesis of indeterminate potential (CHIP) [18,19], where individuals carry a leukemia-related mutation in candidate genes at 2% variant allele frequency but without overt disease [20]. These observations suggest that exposure to smoking may enhance the emergence of mutant HSPC clones.

The continued promotion of E-cigarette safety over combustible cigarettes emphasizes the need to better understand the health-related effects of chronic E-cigarette smoke inhalation. E-cigarette aerosols are not “safe” and consist of ultrafine particles, nicotine, volatile organic compounds, heavy metals such as nickel, formaldehyde, acrolein and flavorings and chronic inhalation of these harmful chemicals predict negative health impacts [21]. E-cigarette exposure can enhance the inflammatory profile of viral and bacterial pathogens of the respiratory system and downregulates innate immunity in the lung [22,23]. Since HSPCs are activated in response to injury and infection, it is likely that E-cigarette exposure has a measurable impact on HSPCs. Because the clonal hematologic disorders MPN and CHIP are associated with smoking history, we evaluated if E-cigarette exposure increases the selective advantage of HSPC carrying MPN and CHIP associated mutations. 

In this study, we used a murine model of chronic nose-only inhalation exposure to assess the effect of E-cigarette aerosol on bone marrow HSPC number and function. We show that chronic E-cigarette exposure negatively impacts HSPC number but not HSC function in transplant assays. Thus, our study findings provide novel evidence on the potential harm of vaping associated products on stem and progenitor cells.

## 2. Results

### 2.1. E-Cigarette Exposure Does Not Impact Peripheral Blood Counts, Bone Marrow Cellularity, or Mature Cells in the Bone Marrow

Chronic conventional cigarette smoke exposure is known to impact peripheral blood cell counts, specifically, it causes an increase in neutrophils and erythrocytes [8,24]. Mice were exposed to nose-only inhalation of E-cigarette smoke and the average particle mass measurement over the study period in the E-cigarette exposure was 88.23 ± 47.01 mg/m^3^. We found that two months of nose-only exposure to nicotine containing E-cigarette smoke did not impact peripheral blood counts (Figure 1A–C) nor did it result in an increased percentage of neutrophils (Figure 1D). Total bone marrow cellularity was not affected by exposure to E-cigarette smoke (Figure 1E). In addition, the frequency and absolute number of lineage positive cells, representing the mature cell population, was equivalent in E-cigarette versus air exposed mice (Figure 1F). Spleen weight was also not impacted by exposure to E-cigarette vapor (Figure 1G). 

### 2.2. E-Cigarette Exposure Suppresses Myeloid Progenitor Populations

Next, we examined the impact of E-cigarette exposure on myeloid progenitor populations including common myeloid progenitors (CMP; Lin^−^c-Kit^+^Sca-1^−^CD34^+^CD16/32^−^), granulocyte-macrophage progenitors (GMP; Lin^−^c-Kit^+^Sca-1^−^CD34^+^CD16/32^+^) and megakaryocyte-erythroid progenitors (MEP; Lin^−^c-Kit^+^Sca-1^−^CD34^−^CD16/32^−^) (Figure 2A). There was no apparent difference in the number of lineage negative (Lin^−^) cells which consists of the immature hematopoietic cells (Figure 2B). However, the number of lineage negative, c-Kit positive (LK; Lin^−^ c-Kit^+^) cells, which is enriched for myeloid progenitors, was significantly reduced in the E-cigarette exposure group when compared with air controls (Figure 2C). Compared to air exposed animals, E-cigarette exposed mice had reduced numbers of CMPs and a trend towards decreased numbers of GMPs, *p* = 0.08, (Figure 2D). However, MEP did not show significant changes in numbers following E-cigarette exposure. These data demonstrate that E-cigarette exposure reduces myeloid progenitors without an appreciable impact on mature myeloid cells.

### 2.3. E-Cigarette Exposure Decreases the Number of Bone Marrow HSPCs

Next, we investigated the impact of E-cigarette exposure on the hematopoietic stem and progenitor compartment (HSPC) using the gating scheme in Figure 3A. E-cigarette exposure decreased the absolute number of HSPCs (*p* = 0.05), defined as lineage negative, c-Kit positive and Sca-1 positive (LKS; Lin^−^ c-kit^+^ Sca-1^+^) surface marker phenotype (Figure 3B). In addition, there was a trend toward decreased long-term repopulating hematopoietic stem cells (LT-HSCs), defined as (LKS-SLAM; LKS-CD48^−^ CD150^+^), but failed to reach significance (*p* = 0.14) (Figure 3C). This observed decrease in absolute number of LKS cells, which contains LT-HSCs, short-term HSCs (ST-HSCs) and multipotent progenitors (MPPs) as well as downstream myeloid committed progenitors, suggests that E-cigarette exposure leads to dysregulation of bone marrow hematopoiesis.

### 2.4. HSC Function is Unperturbed Following E-Cigarette Smoke Exposure

To investigate the effects of E-cigarette exposure on hematopoietic stem and progenitor cell function we performed competitive transplantation experiments. We transplanted lethally irradiated recipient mice with whole bone marrow cells from either air or E-cigarette (CD45.2) exposed mice along with unexposed wildtype competitor cells (CD45.1), and then peripheral blood was analyzed for chimerism each month post-transplant (Figure 4A). Air and E-cigarette exposed cells showed comparable contribution to peripheral blood leukocytes over time (Figure 4B). Peripheral blood cell counts were also assessed, though we found no differences in leukocyte, erythrocyte and platelet counts (Figure 4C–E) in air versus E-cigarette exposed mice. Thus, these data suggest that E-cigarette exposure does not markedly impact HSPC competitive fitness or functional output.

### 2.5. E-Cigarette Exposure Does Not Impair HSC Engraftment in Response to LPS

To further determine the functional relevance of reduced HSPC and myeloid progenitor populations in E-cigarette exposed mice, we assessed HSC function of air and E-cigarette exposed mice in response to an acute inflammatory challenge using the Gram-negative bacterial component lipopolysaccharide (LPS). An acute low-dose LPS challenge is known to initiate emergency myelopoiesis through proliferation of LT-HSCs and MPPs to increase the output of innate immune cells [25,26] but it does not alter their long-term reconstitution potential [27]. In this regard, mice exposed to air or E-cigarette for two months were subsequently challenged with a single dose of LPS (35 µg/mouse) 16 h before euthanasia. Since LPS causes significant systemic effects, we assessed change in body weight post-LPS treatment and found no difference between air and E-cigarette exposure groups (Appendix A). As expected, we observed an expansion of LKS, LKS-SLAM and myeloid progenitors in all LPS treated mice (Appendix A), yet there were no significant differences in the absolute numbers of these populations between air and E-cigarette exposures. Whole bone marrow was then used for competitive repopulation assays, and peripheral blood chimerism was analyzed by flow cytometry each month (Figure 5A). Bone marrow from E-cigarette exposed animals showed comparable reconstitution to air exposed controls (Figure 5B). These data demonstrate that two months of E-cigarette exposure followed by an acute inflammatory stimulus does not affect HSC recovery in primary transplants. However, the E-cig+LPS group demonstrated a small reduction in hematopoietic recovery at all time points compared to the Air+LPS group and the significance of this effect is most likely masked by the variance contribution from animal to animal variation. At 5 months post-transplant, mice that received bone marrow from E-cigarette exposed LPS challenged mice demonstrated a marked increase in peripheral blood monocyte counts (Figure 5C). However, total leukocytes, CD45.2 derived (air or E-cigarette exposed) leukocytes, red blood cells, and platelet counts remained unaltered in E-cigarette group compared to controls (Figure 5D–G). Thus, HSCs subject to an inflammatory challenge from E-cigarette exposed mice showed a persistent effect of the LPS challenge several months post-transplant unlike the air controls.

### 2.6. JAK2^V617F^ Mutant Cells Gain an Early Selective Advantage Following E-Cigarette Exposure

Chronic inflammation is a hallmark feature in myeloproliferative neoplasms (MPNs) and is a driver of disease progression. Data indicates that inflammatory signals such as LPS influence the selective outgrowth and expansion of myeloid leukemia associated somatic mutations such as *JAK2^V617F^* and *TET2*^−/−^ [28,29]. Therefore, we sought to determine whether E-cigarette exposure could promote the selective expansion of mutant cells utilizing competitive transplants of *JAK2^V617F^* mutated and *TET2* knockout bone marrow cells. (Figure 6A). After one month of E-cigarette exposure, the relative frequency of *JAK2^V617F^* cells in the peripheral blood was almost significantly more than the air exposed group (*p* = 0.058) (Figure 6B). However, at the end of two months of exposure, both air and E-cigarette groups showed similar levels of *JAK2^V617F^* cells in the blood (Figure 6B). In contrast, *TET2* knockout cells were not affected by E-cigarette exposure compared to the control group (Figure 6C). To determine if E-cigarette exposure selects for mutant cells in the bone marrow, we measured the percentage of *JAK2^V617F^* cells in each of the HSPC sub-types at sacrifice. We found no difference between control and E-cigarette exposure in the number of *JAK2^V617F^* mutant HSPCs in the bone marrow (Figure 6D).

## 3. Discussion

We have found that chronic exposure to E-cigarette smoke can alter hematopoietic stem and progenitor populations in the bone marrow. Specifically, a two-month E-cigarette exposure decreased the absolute numbers of hematopoietic myeloid progenitors (LK; Lin^−^ c-Kit^+^) cells as well as CMP and GMP. However, there was no appreciable effect on the reconstitution potential of whole bone marrow from E-cigarette exposed mice or from E-cigarette exposed mice combined with an inflammatory challenge such as LPS in competitive repopulation transplant assays. Further, mice transplanted with bone marrow from donors that had been exposed to E-cigarettes plus LPS had increased monocytes in the peripheral blood that persisted at 20 weeks post-transplant, mimicking an “aged” HSC phenotype.

Exposure of mice to conventional cigarette smoke induces bone marrow myelosuppression with significant reductions in the phenotypic LKS (HSPC) population [9,30,31,32]. However, there is currently no evidence on the pathological changes, if any, in bone marrow HSPCs following E-cigarette exposure. It is notable that E-cigarette exposure leads to a decline in LKS numbers, implying a potential harmful effect on hematopoiesis in those who use E-cigarettes. It is possible that the effects we are seeing are nicotine-dependent, prior reports suggest that sustained nicotine exposure leads to bone marrow suppression and extramedullary hematopoiesis in the spleen [33]. Interestingly, the suppressive effect of E-cigarette exposure on myeloid progenitors, which are the immediate precursors to differentiated cells, did not produce changes in peripheral blood cell counts. This phenomenon is similar to the hematological sequalae of conventional cigarette smoking. Although chronic smokers have increased peripheral blood leukocyte counts and neutrophilia [24], bone marrow myelosuppression has been observed in mice exposed to chronic cigarette smoke for nine months [31]. This was attributed to dysfunctional mesenchymal stem cells in the bone marrow stem cell niche leading to abnormal hematopoiesis [31].

Our study indicates that despite the reduction in phenotypic LKS and myeloid progenitors, competitive reconstitution using whole bone marrow was not negatively affected in primary transplants. Given that we did not observe a highly significant decline in the LT-HSC population in the E-cigarette exposure group, it is most likely that the subtle difference was not sufficient to induce a difference in reconstitution capacity. In addition, phenotypic changes in HSPC populations may not necessarily reflect HSC functionality. In addition, we did not assess the long-term HSC reconstitution ability in secondary and tertiary transplants and this would need to be characterized in the future to determine if E-cigarette exposure has an impact on HSC self-renewal.

The continuous output of mature blood and immune cells is maintained by self-renewing HSCs to maintain homeostasis. However, during infection induced hematologic stress HSCs undergo active cell cycling to increase production of blood and immune cells to replace consumed effector cells [6]. These changes in HSPC numbers and activity are mediated by the presence of pathogen-associated molecular patterns (PAMP) receptors such as Toll-like receptor 4 (TLR4) for LPS on the surface of HSCs that can directly affect HSC signaling cascades [34] and also the release of inflammatory cytokines that trigger HSC division and differentiation [35,36,37]. Paradoxically, increased stem cell division is associated with impaired self-renewal that eventually leads to the depletion of the HSC pool [38]. Several studies have demonstrated that chronic exposure to LPS decreases HSC competitive fitness and hematopoietic reconstitution [39,40,41] by direct TLR4 activation [34]. More recently, Mann et al., 2018, demonstrated that a single acute exposure of LT-HSCs to low dose LPS in vitro did not affect the long-term reconstitution of hematopoietic cells in primary and secondary transplanted mice [27]. However, LPS challenged LT-HSCs from old mice (aged HSCs) displayed an increased myeloid cell output on top of the expected age-dependent myeloid bias [42,43], which was not observed in HSCs from young mice [27]. A “memory” of the initial inflammatory challenge has been proposed to be at the helm of this myeloid skewing [27]. E-cigarette exposed mice challenged with acute LPS did not demonstrate decreased reconstitution compared to the air group post-transplant with the caveat being the absence of secondary transplant data to rigorously assess long-term reconstitution. Mice transplanted with bone marrow from E-cigarettes plus LPS donors had elevated monocytes compared to mice transplanted with bone marrow from air plus LPS donors, indicating a long-term myeloid skewing akin to responses by aged HSCs. This indicates that chronic E-cigarette exposure can induce an aged HSC phenotype. Further studies are required to address the specific gene signature changes that occur in E-cigarette exposed HSCs that lead to an aged phenotype. 

Inflammation is implicated as an environmental driver of myeloid malignancy. Smoking is a source of chronic inflammation and is strongly associated with clonal hematopoiesis and CHIP [18,19]. Although E-cigarette aerosols provoke local inflammation in the airways and lungs [44], systemic inflammation has not been reported. To investigate the impact of E-cigarette exposure in the selection and expansion of pre-leukemic HSPCs, we determined the extent to which two months of E-cigarette exposure influenced the outgrowth of *JAK2^V617F^* mutated and TET2 knockout HSPCs. We did observe a transient increase in *JAK2^V617F^* cells in the blood after one month of E-cigarette exposure but not after two months, possibly due to a nicotine tolerance effect. At the end of the two-month exposure period, both air and E-cigarette exposure groups displayed increase *JAK2^V617F^* chimerism relative to baseline at eight weeks post-transplant. This was interesting, since this knock-in mouse model of *JAK2^V617F^* does not confer HSPCs with a significant competitive advantage in transplant assays and shows progressively decreasing levels of mutant cells over time [45]. The fact that both control and treatment groups display a sudden increased mutant chimerism indicates a common underlying cause which we hypothesize is stress-related arising from the nose-only exposure system. Since it has been established that MPN physical symptom burden is associated with psychological stress [46], it would be interesting to evaluate psychologic stress on MPN disease progression. Mutations in *TET2* is one of the commonly occurring pre-leukemic mutations in CHIP and exerts clonal dominance by increased self-renewal under inflammatory conditions. *TET2* knockout cells did not show a selective growth advantage under the current E-cigarette exposure conditions in concordance with recent findings that *TET2* mutants are not significantly favored by smoking status [47].

Thus, while E-cigarette exposed mice presented with normal hematopoietic reconstitution under steady state as well as following acute systemic inflammation with LPS challenge, increased myeloid bias was observed at late time points post-inflammatory challenge. These observations demonstrate that not only do inhaled aerosols from E-cigarette smoke disrupt the bone marrow hematopoietic stem cell pool, they also induce myeloid skewing in response to an acute inflammatory stimulus. Given the extensive number of adolescents using E-cigarettes on a consistent basis, these alterations in HSC numbers demonstrated in our E-cigarette exposure model, highlight the potential risk for bone marrow dysfunction that chronic E-cigarette use may pose.

## 4. Materials and Methods 

### 4.1. Mice

All animal procedures were performed under the approval of the Institutional Animal Care and Use Committee at the University of California, Irvine. Eight-week-old female C57BL/6J (CD45.2) mice were purchased from The Jackson Laboratory (JAX stock #000664) and housed under specific pathogen–free conditions at the University of California, Irvine. The *JAK2^V617F^* conditional knock-in mouse model has been previously described [45] and was crossed with Vav-Cre mice to induce expression of the mutant allele in hematopoietic cells. The *TET2* knockout (*TET2*^−/−^) mouse model has been described [48] and heterozygous breeding pair was purchased from The Jackson Laboratory (JAX stock #023359).

### 4.2. E-Cigarette Exposure in Mice

We used the commercially available Vibe electronic cigarette and e-liquid from VaporFi in our lab-made exposure apparatus. The system consists of an actuator, battery, which provides 70 Watts of power, peristaltic pump, dampening chamber, and nose-only exposure chamber. The E-cigarette contains an atomizer with a 316 L stainless steel coil and a tank with a 3.5 mL liquid capacity. E-liquids were tobacco flavored and consisted of a 50:50 ratio of propylene glycol and vegetable glycerin. Nicotine (Acros Organics) was added at a final concentration of 15 mg/mL. Each cycle of E-cigarette smoke exposure provided a 55 mL puff volume over 2 s with an inter-puff interval of 28 s. The E-cigarette vapor had a final average particle concentration measured at the nose-only port of 88.23 +/− 47.01 mg/m^3^. Mice were exposed nose-only [49,50] for 2 h/day, 4 days/week, for 2 months.

### 4.3. Peripheral Blood Cell Counts

Peripheral blood was obtained from the saphenous vein and hematologic parameters were analyzed by an automated cell counter machine (ABCVet Hemalyzer, scil). 

### 4.4. Flow Cytometry of Bone Marrow Cell Populations

All antibodies were purchased from BioLegend. Whole bone marrow (BM) was harvested from both femurs and tibiae, RBCs were lysed with ACK buffer and cells were stained for 30 min on ice. Pacific blue conjugated antibodies against Ter119, Mac1, Gr1, B220, CD3, CD4 and CD8 were used to detect mature cells (Lineage cocktail). HSC populations were detected by staining with mouse specific antibodies against Lineage markers, c-Kit, Sca-1 (LKS), CD48 and CD150 (LKS-SLAM). Progenitor subsets (CMP/GMP/MEP) were detected using antibodies against CD34 and CD16/32. Flow cytometry was performed on the Novocyte (ACEA Biosciences) at the UCI Immunology Core facility. Data were analyzed using FlowJo software (FlowJo^TM^ software, BD Biosciences, San Jose, CA, USA).

### 4.5. Competitive Bone Marrow Transplant and Peripheral Blood Chimerism 

Bone marrow was harvested from donor mice (CD45.2) exposed to either air or vape B6.SJL-Ptprca Pepcb/Boy (BoyJ, CD45.1) mice were purchased from The Jackson Laboratory. Recipient mice (CD45.1/CD45.2) were lethally irradiated at 8 Gy the day before transplantation. Donor whole bone marrow cells (CD45.2 air or E-cigarette) were mixed in a ratio of 1:1 (1 × 10^6^ each) with competitor cells (CD45.1) and injected intraveneously (retro-orbital) into irradiated recipients. To develop chimeric mice containing *JAK2^V617F^* mutant cells, whole bone marrow cells from *JAK2^V617F^* knock-in mice were mixed in a 1:1 ratio with wildtype competitor cells and injected into lethally irradiated recipients. To develop a mouse model of clonal hematopoiesis of indeterminate potential (CHIP), we generated bone marrow chimeras of mice carrying a small percentage of *TET2*^−/−^ cells by injecting *TET2*^−/−^ cells and competitor (wild-type) cells in a ratio of 1:99. Mice were rested for 4 to 8 weeks before beginning E-cigarette exposure for 2 months. The repopulating advantage of *JAK2^V617F^* and *TET2*^−/−^ cells upon E-cigarette exposure was followed for 2 months. Flow cytometry for chimerism of mature cells in peripheral blood was performed using CD45.1 and CD45.2 antibodies. Data was acquired on the BD Accuri C6 (BD Biosciences, San Jose, USA).

### 4.6. Statistical Analysis 

Data are shown as mean and error bars indicate the standard error of mean (SEM). *p* value was calculated using an unpaired *t*-test comparing the means of two groups (GraphPad Software, Prism version 8.3.1, San Diego, CA, USA).

## 5. Conclusions

The present study shows, for the first time, that continuous E-cigarette exposure promotes bone marrow myelosuppression by decreasing the numbers of HSPCs and myeloid progenitor subtypes. Our findings suggest that E-cigarette exposure combined with an acute inflammatory challenge can lead to delayed monocytosis. Overall, the modulation of the bone marrow HSPC pool by E-cigarette exposure represent the potential negative health-effects of E-cigarette use.

## Figures and Tables

**Figure 1 cancers-12-02292-f001:**
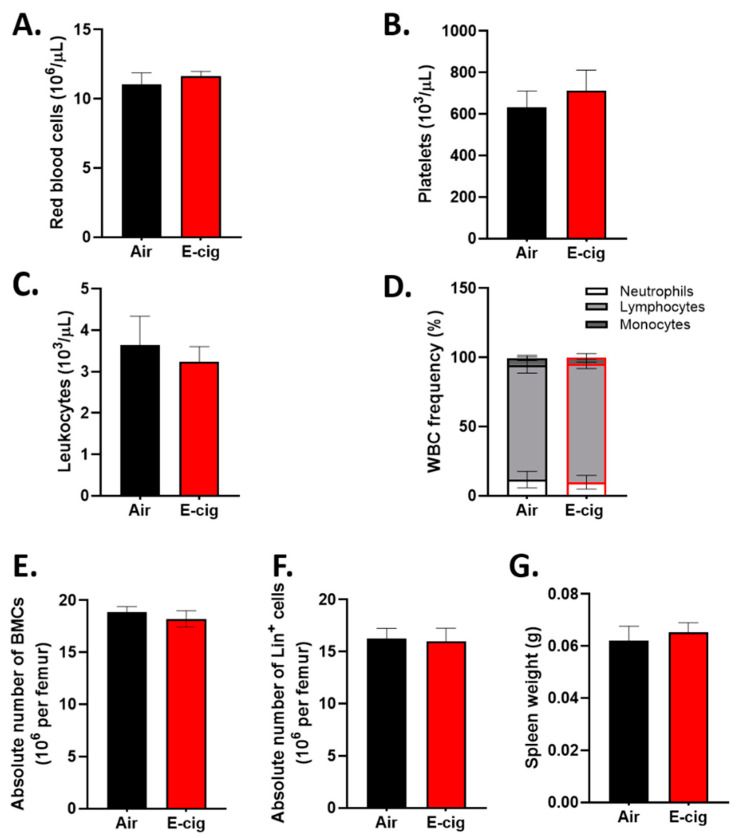
Two-month E-cigarette exposure does not affect peripheral blood cell counts of (**A**) red blood cells, (**B**) platelets, (**C**) leukocytes, and (**D**) leukocyte differentials in C57Bl/6J mice. (**E**) Bone marrow cellularity, (**F**) lineage positive cells, and (**G**) spleen weight after E-cigarette exposure in air and E-cigarette exposed mice. Data are shown as mean ± SEM, *n* = 4–7 mice/group. WBC, white blood cell; BMCs, bone marrow cells; Lin^+^, lineage positive.

**Figure 2 cancers-12-02292-f002:**
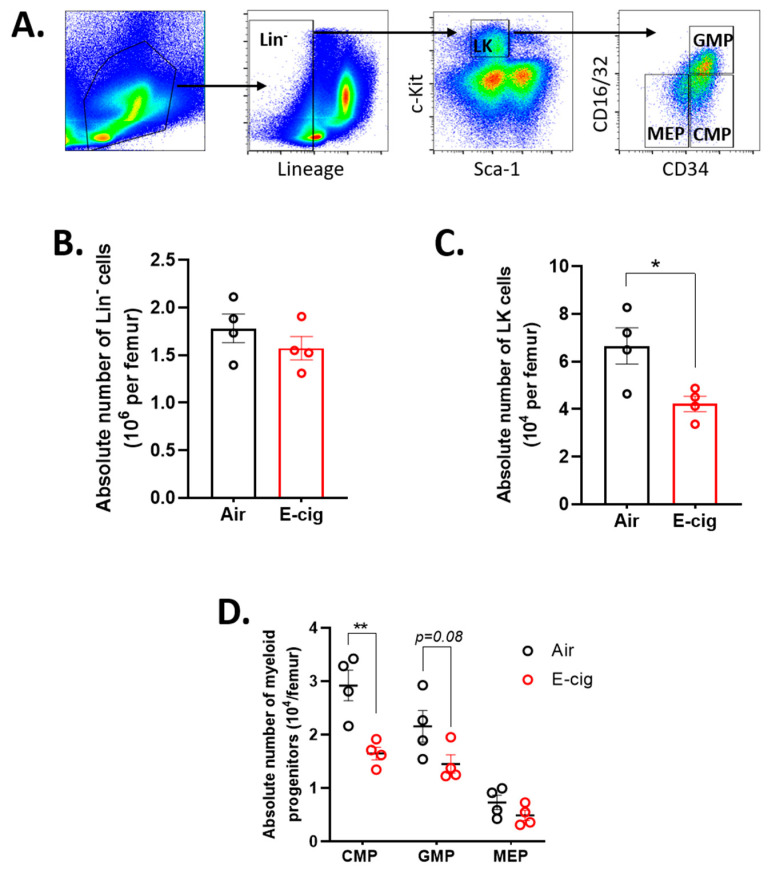
E-cigarette exposure suppresses myeloid progenitor cell numbers in the bone marrow. (**A**) Gating scheme for the characterization of progenitor cell sub-types. Absolute numbers of (**B**) lineage- (Lin-), (**C**) lineage-, c-Kit+ (LK) and (**D**) CMP, GMP and MEP per femur. Data are shown as mean ± SEM, *n* = 4 mice/group. * *p* < 0.05, ** *p* < 0.01, unpaired student’s *t*-test.

**Figure 3 cancers-12-02292-f003:**
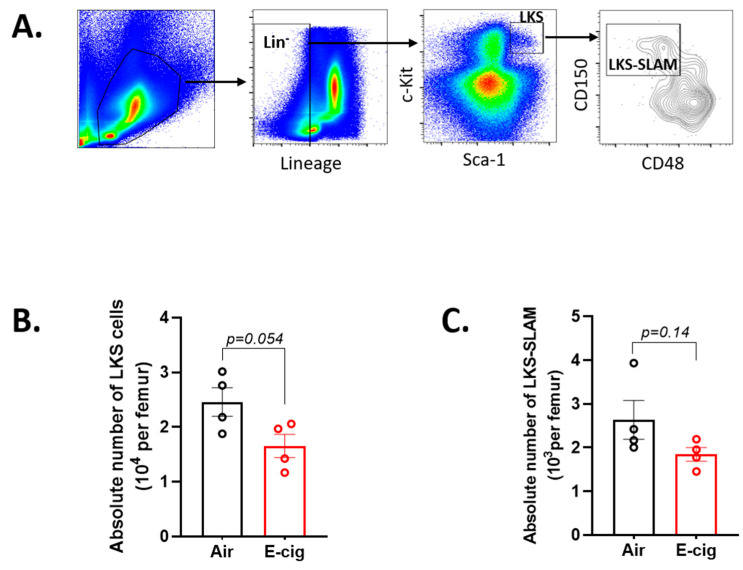
E-cigarette exposure suppresses HSPC numbers in the bone marrow. (**A**) Gating scheme for the characterization of HSPC sub-types. Absolute numbers of (**B**) LKS and (**C**) LKS-SLAM populations per femur. Data are shown as mean ± SEM, *n* = 4 mice/group. *p*-value determined by unpaired student’s *t*-test.

**Figure 4 cancers-12-02292-f004:**
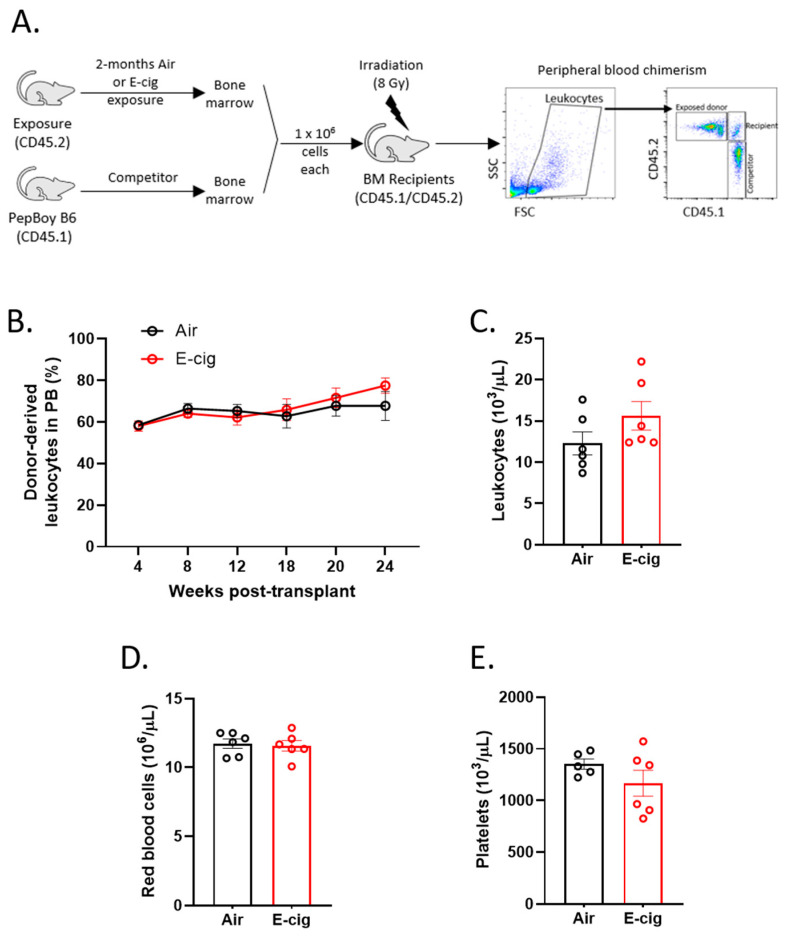
HSC function is unperturbed following E-cigarette exposure. (**A**) Competitive transplant set-up and peripheral blood (PB) chimerism analysis. (**B**) PB chimerism of CD45.2 leukocytes post- transplant. Peripheral blood cell counts of (**C**) leukocytes, (**D**) red blood cells and (**E**) platelets at 20 weeks post-transplant. Data are shown as mean ± SEM, *n* = 5–6 mice/group. BM, bone marrow.

**Figure 5 cancers-12-02292-f005:**
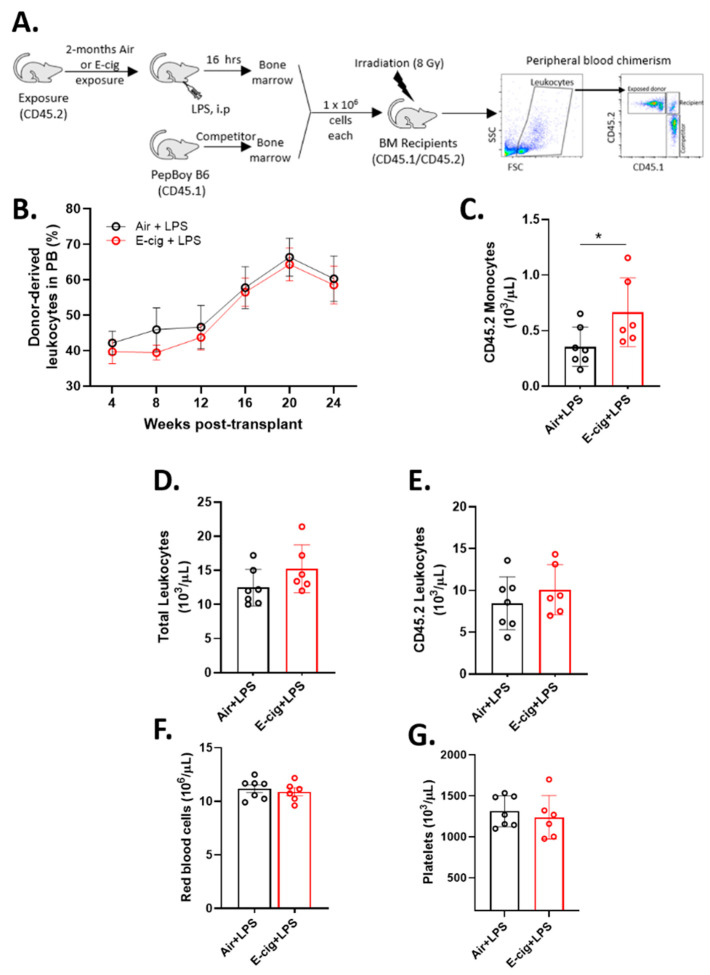
Systemic LPS post-E-cigarette exposure results in persistent monocytosis. (**A**) Experimental design. (**B**) Peripheral blood chimerism of CD45.2 leukocytes from air and E-cigarette exposed donor cells. Peripheral blood counts of (**C**) CD45.2 monocytes, (**D**) leukocytes, (**E**) CD45.2 leukocytes, (**F**) red blood cells, and (**G**) platelets 20 weeks post-transplant. Data are shown as mean ± SEM. *n* = 5–6 mice/group. * *p* < 0.05, unpaired student’s *t*-test.

**Figure 6 cancers-12-02292-f006:**
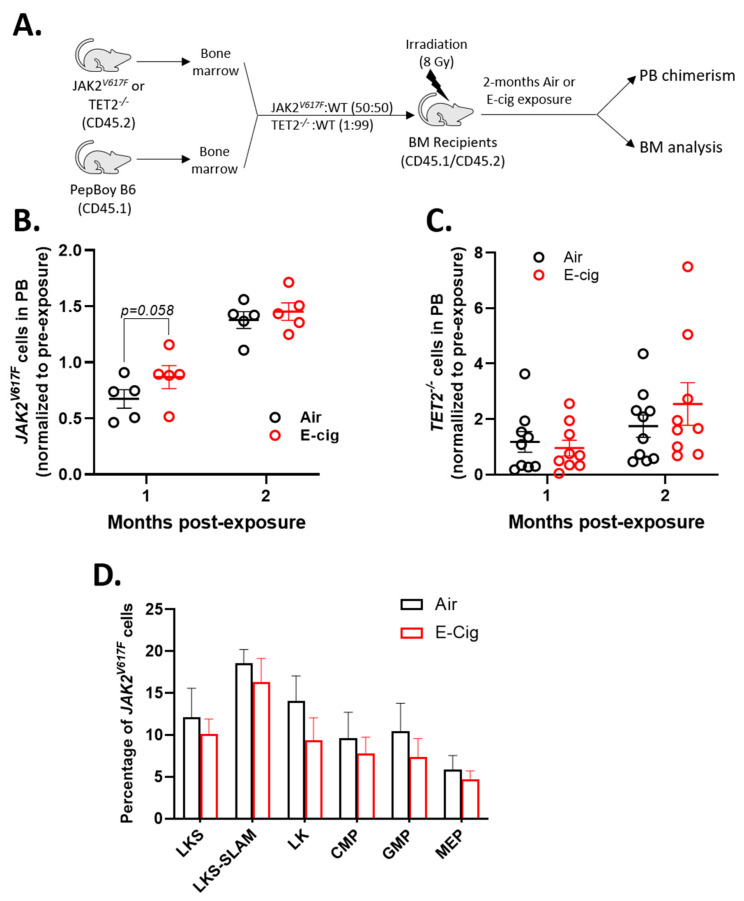
E-cigarette exposure increases peripheral blood chimerism of *JAK2^V617F^* mutant cells. (**A**) Experimental design. Relative change in peripheral blood chimerism of (**B**) *JAK2^V617F^* mutant and (**C**) *TET2*^−/−^ cells from pre-exposure. (**D**) Relative contribution of *JAK2^V617F^* to HSPC populations after 2 months of exposure. Data are shown as mean ± SEM. *n* = 5–10 mice/group.

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
