# Peer review of "E-Cigarette Exposure Decreases Bone Marrow Hematopoietic Progenitor Cells"

_cancers, 2020, doi:10.3390/cancers12082292_

Round 1

Reviewer 1 Report

The authors used mice study showing E-cigarette exposure promotes bone
marrow myelosuppression by decreasing the numbers of HSPCs and myeloid progenitor subtypes. Further study also suggest that E-cigarette exposure combined with an acute inflammatory challenge impairs hematopoietic stem cell reconstitution at early time points. Though the finding is interesting, there are still some questions on this manuscript.

  1. The authors show E-cigarette could damage stem cell in number, how about the long term effect? Could the marrow recover or not in a long run?
  2. In transplant model, the author compare E-cigarette group to air-exposure group, how about these group compare to the normal mice control?
  3. Is there any possible mechanism for the damage? Or is there possible potential substances for the damage, like nicotine?  If so, could there any further study to proof the theory?
  4. Is the effects works the same on young mice and the old mice?
  5. The results is not good for publish in the journal. The authors should provide more evidences or mechanisms to the paper.

Author Response

The authors used mice study showing E-cigarette exposure promotes bone marrow myelosuppression by decreasing the numbers of HSPCs and myeloid progenitor subtypes. Further study also suggest that E-cigarette exposure combined with an acute inflammatory challenge impairs hematopoietic stem cell reconstitution at early time points. Though the finding is interesting, there are still some questions on this manuscript.

  1. Reviewer’s comment

The authors show E-cigarette could damage stem cell in number, how about the long-term effect? Could the marrow recover or not in a long run?

Author’s response

We agree that continued exposures, > two months, are required to assess the long-term effects of chronic E-cigarette inhalation on bone marrow hematopoietic stem and progenitor cells (HSPCs). However, currently, there are no reports on the effects of E-cigarette exposure on bone marrow HSPCs. Our current study was undertaken to fill this knowledge gap.

Previous work with combustible cigarette smoke exposure in mice demonstrated that both 7 weeks exposure (Xie, 2014, cigarette myelosuppression TSG-6) and an extended period of 9 months exposure (Siggins 2014,hematopoietic stem cell niche) resulted in a similar decrease, 50-60% decrease, in the HSPC population, i.e. the lineage negative, c-Kit positive, Sca-1 positive (LKS) cells. Thus, we speculate that prolonged E-cigarette exposure will continue to exert a similar level of myelosuppression as observed with 2 months exposure.

In order to determine how the function of hematopoietic stem cells is affected, we performed transplant assays with Air and E-cigarette exposed bone marrow cells and did not observe any difference in stem cell recovery (Figure 4B, main text). However, long-term re-constitutive potential can only be determined from secondary and tertiary transplants. On the other hand, it is possible that longer E-cigarette exposures, > two months, could have a negative impact on HSC function in primary transplants. These questions need to be addressed in future studies. 

  1. Reviewer’s comment

In transplant model, the author compare E-cigarette group to air-exposure group, how about these group compare to the normal mice control?

Author’s response

Thank you for this comment. Because air-exposed animals are subject to the stress of being restrained daily, it is also important to make a comparison to control unmanipulated mice. In the transplant model, we used standard CD45.1 competitor bone marrow to determine how Air or E-cigarette exposure impacts engraftment in lethally irradiated mice. We also transplanted unexposed CD45.2 bone marrow cells along with CD45.1 competitor bone marrow as controls. In the figure below, we show that Air exposed, E-cigarette exposed, and unexposed wild-type controls show similar engraftment post-transplant.

Figure 1. HSC engraftment is not impacted by air or E-cigarette exposure. Competitive bone marrow transplant was performed with CD45.2 donor bone marrow cells from Air exposed, E-cigarette exposed and unexposed wildtype mice along with CD45.1 competitor cells. Peripheral blood (PB) chimerism was determined at regular intervals post-transplant to evaluate hematopoietic reconstitution.

  1. Reviewer’s comment

Is there any possible mechanism for the damage? Or is there possible potential substances for the damage, like nicotine?  If so, could there any further study to proof the theory?

Author’s response

E-liquids are composed of nicotine and its associated solvents, propylene glycol and vegetable glycerin along with flavoring chemicals to make them more appealing to youth users. Toxicological studies have also highlighted detectable levels of heavy metals, formaldehyde and acrolein in the vapor of some E-cigarette products (Logue, 2017, emissions from electronic cigarettes).  

Our exposure system utilized E-liquid containing nicotine and we hypothesize that the effects observed in the bone marrow hematopoietic cells are nicotine-dependent. Interestingly, a recent publication also demonstrates that a negative inhibitor of the α7 nicotinic acetylcholine receptor (α7nAChR) increases the bone marrow hematopoietic stem cells reservoir (Costantini 2019, uniquely human hematopoietic stem cell) indicating that chronic α7nAChR activation has the opposite effect.

Current studies are aimed at conducting exposures using E-liquid without nicotine but beyond the scope of this current paper. Another future direction is the use α7nAChR knockout mice in exposures.

  1. Reviewer’s comment

Is the effects works the same on young mice and the old mice?

Author’s response

We agree that investigating any disparity in the effects of E-cigarette exposure between young and old mice would be important, given that the most concerning aspect of E-cigarette usage is its popularity among adolescents and young adults. However, in this study, we have only determined the effects of E-cigarette exposure on young mice. Since old mice already display hematopoietic defects such as myeloid bias and decreased self-renewal potential (Weissman 2005, stem cell aging), we speculate that the effects of E-cigarette exposure will be more pronounced in old mice. However, repeating our experiments with old mice is beyond the scope of this current manuscript.

  1. Reviewer’s comment

The results is not good for publish in the journal. The authors should provide more evidences or mechanisms to the paper.

Author’s response

We acknowledge that additional mechanistic evidence will significantly improve the manuscript. However, because so little is known about the impact of E-cigarette exposure on hematopoiesis, we believe our initial observations are important to report as soon as possible with more mechanism studies to come in future reports.

Reviewer 2 Report

The very elegant work, clearly indicating that chronic exposure of animals to the E-cigarettes results  in decreased in common myeloid progenitors and granulocyte-macrophage progenitors and impairment of  short-term reconstitution when exposed to an additional inflammatory challenge compared to unexposed animals.

Author Response

Reviewer #2:

The very elegant work, clearly indicating that chronic exposure of animals to the E-cigarettes results in decreased in common myeloid progenitors and granulocyte-macrophage progenitors and impairment of short-term reconstitution when exposed to an additional inflammatory challenge compared to unexposed animals.

In this paper, Ramanathan et al tested if chronic inhalation of Electronic cigarette (E-cigarette) aerosol influences adult hematopoiesis in a mouse model. The authors show that chronic E-cigarette exposure negatively impacts the number of hematopoietic stem progenitor cells, particularly myeloid progenitors, in the bone marrow. They also claim that E-cigarette exposure with LPS-induced systemic inflammation impairs short-term reconstitution activity and causes monocytosis in transplanted recipients. Furthermore, using JAK2 V617F-mutated bone marrow cells, they show that one month of E-cigarette exposure slightly increased the mutant cells.

Author’s response

We thank the reviewer for the close reading of our paper and supportive comments. We are pleased that the reviewer is satisfied with our manuscript. 

Reviewer 3 Report

In this paper, Ramanathan et al tested if chronic inhalation of Electronic cigarette (E-cigarette) aerosol influences adult hematopoiesis in a mouse model. The authors show that chronic E-cigarette exposure negatively impacts the number of hematopoietic stem progenitor cells, particularly myeloid progenitors, in the bone marrow. They also claim that E-cigarette exposure with LPS-induced systemic inflammation impairs short-term reconstitution activity and causes monocytosis in transplanted recipients. Furthermore, using JAK2 V617F-mutated bone marrow cells, they show that one month of E-cigarette exposure slightly increased the mutant cells.

Given that E-cigarettes are quickly spreading in the world as an alternative to conventional tobacco smoking, I understand the importance of this study. However, as the data of transplantation experiments only show very subtle difference, I am afraid that the authors’ claims written in the abstract are not sufficiently supported by their observation. In Figure 5B and C, the statistical evaluation should be performed by comparing Air+LPS and E-cigarette groups.

Author Response

Reviewer #3:

Given that E-cigarettes are quickly spreading in the world as an alternative to conventional tobacco smoking, I understand the importance of this study. However, as the data of transplantation experiments only show very subtle difference, I am afraid that the authors’ claims written in the abstract are not sufficiently supported by their observation. In Figure 5B and C, the statistical evaluation should be performed by comparing Air+LPS and E-cigarette groups.

Author’s response

We have modified the abstract, lines 20-23 and 30-32, to reflect the subtle differences observed in the competitive transplantation experiments. 

Competitive reconstitution in bone marrow transplants was not affected by two months of E-cigarette exposure but when these mice were challenged with an additional inflammatory stimulus using lipopolysaccharide (LPS), competitive fitness was slightly reduced at early time points post-transplant (≤8 weeks).

Altogether, our findings reveal that chronic E-cigarette exposure alters the bone marrow HSPC populations and moderately impairs short-term competitive reconstitution when exposed to an additional inflammatory challenge.

We have also altered the title to “E-cigarette Exposure Decreases Bone Marrow Hematopoietic Progenitor Cells and Impairs Short-term Reconstitution in the Setting of Acute Inflammatory Stimulus” to remove the claim the E-cigarettes impact hematopoietic stem cells.

The competitive transplant setting provides more sensitivity to detect moderate differences in stem cell fitness. Although we observe a slight decrease in competitive fitness, we understand that the fitness of E-cigarette exposed and LPS challenged HSCs will not be defective in a non-competitive setting. Since the effects of E-cigarette exposure on HSC fitness are not substantial, we have described the transplant data as fitness of HSCs when compared to a standard competitor and performed statistical tests using this comparator.

Round 2

Reviewer 1 Report

The authors responsed to the questions and modified the manuscript. I agree that the article is good to be published in the journal.

Author Response

See responses to Reviewer #3 word doc.

Reviewer 3 Report

In the response letter, the authors answered to my last comment “In Figure 5B and C, the statistical evaluation should be performed by comparing Air+LPS and E-cigarette+LPS groups”

as below;

The competitive transplant setting provides more sensitivity to detect moderate differences in stem cell fitness. Although we observe a slight decrease in competitive fitness, we understand that the fitness of E-cigarette exposed and LPS challenged HSCs will not be defective in a non-competitive setting. Since the effects of E-cigarette exposure on HSC fitness are not substantial, we have described the transplant data as fitness of HSCs when compared to a standard competitor and performed statistical tests using this comparator.

I understand what the authors want to tell, but I am afraid that this flaw might have the readers cast doubts on all the story in this paper. The authors admit that E-cigarette exposure + LPS has no substantial effects on HSC fitness whereas they are describing the substantial inferior reconstitution of E-cigarette exposure + LPS bone marrow cells by comparing CD45.1+ competitor bone marrow cells in Figure 5C. In the experimental setting, the right control for “E-cigarette exposure + LPS bone marrow cells” is “Air exposure + LPS bone marrow cells”. Therefore, the results shown in the current Figure 5B and 5C are not based on true science and seem to draw arbitrary interpretation. I strongly recommend that Figure 5B and 5C be appropriately revised or just removed.
